# Can Government Environmental Auditing Help to Improve Environmental Quality? Evidence from China

**DOI:** 10.3390/ijerph20042770

**Published:** 2023-02-04

**Authors:** Xiaoyu Li, Jiawei Tang, Chao Feng, Yexiao Chen

**Affiliations:** 1School of Accounting, Chongqing Technology and Business University, Chongqing 400067, China; 2Research Center for Economy of Upper Reaches of the Yangtze River, Chongqing Technology and Business University, Chongqing 400067, China; 3School of Economics and Business Administration, Chongqing University, Chongqing 400030, China

**Keywords:** environmental auditing, environmental quality, government competition, financial situation, institutional environment

## Abstract

Promoting environmental governance to achieve green and low-carbon development is the focus of attention from all walks of life. As a policy tool to control environmental pollution, the effectiveness of environmental audits remains to be verified. Based on China’s provincial data from 2004 to 2019, this paper aims to examine the impact and mechanisms of government environmental auditing on environmental quality. Results show that government environmental auditing helps to improve overall environmental quality, but there is a certain lag effect occurring. The heterogeneity test suggests that the impact of environmental auditing on comprehensive environmental quality is more significant when the government competition is smaller, the financial situation is better, and the institutional environment is weaker. Our analysis provides empirical evidence for understanding the role and function of government environmental auditing in environmental governance.

## 1. Introduction

Industrialization has until now helped many societies and their economies progress but it has also brought about what is known as the “silent spring” effect. At the same time, “the limit of growth” calls for people to reconsider how economic development should proceed. After decades of growth since the reforms of the late 1970s and opening the country to the world, China is now the second-largest economy, the largest manufacturer, the largest trader of goods, and the second-largest consumer of goods. Despite the rapid growth of China’s economy, it has paid a huge price in the form of a damaged or heavily polluted environment and resources [1]. Taking air pollution as an example, in 2015 the Ministry of Environmental Protection implemented its air quality forecast and warnings for the key cities nationwide, and released real-time monitoring data on air quality. According to the State of the Environment in China 2015*,* of the 338 cities at or above the prefectural level, 265 exceeded the environmental air quality standard, accounting for 78.4%. Despite significant progress being made in the battle against pollution in recent years, according to the State of the Environment in China 2019*,* the air quality in 180 of the country’s 337 cities exceeded the standard in 2019, accounting for more than half of the cities.

The national environmental authority endeavors to develop more Innovative approaches that encourage corporations to conduct better environmental management. To improve environmental management, China explores ways to disclose corporate environmental information [2]. However, the selectively disclosing status of corporate environmental information disclosure is significantly different across different industrial sectors, company size, and company ownership [3]. Moreover, there are strong political connections present between the state and firms’ management in China [4]. Government intervention and political connections are influencing corporations’ performance [5,6], firm value [7], social responsibility [8,9], the quality of corporate social responsibility disclosure [10], innovation [11,12,13], and so on.

Meanwhile, corporate environmental strategies are also significantly influenced by government pressures [14]; the local government’s emphasis on the scarcity of natural environmental resources and the importance of eco-environmental protection puts corporations under external supervision, and external pressure will prompt corporations to change their environmental strategies and promote corporate green development [15,16].

Consequently, environmental governance plays a critical role to promote regional green transformation [17]. Local governments in China assume responsibility for environmental governance in response to environmental policies implemented by the central government to promote regional green and sustainable development [18]. Additionally, the Chinese government has established a vertical supervision mechanism represented by the Central Environmental Protection Inspection (CEPI), to directly regulate their environmental governance through monitoring local governments at all levels [19]. It actually enhanced environmental quality [20], and effectively promoted green transformation regionally by reducing local pollution emissions and improving total factor productivity [21].

Despite the important role of auditing in national governance matters, China’s environmental auditing started very late. In 2014, some regions carried out exploratory trials to audit outgoing officials’ natural resource management. In 2015, the first government environmental auditing pilot work was carried out in Gansu. In 2017, the auditing of outgoing officials’ natural resource management was formally approved and implemented. Although existing studies believe that environmental auditing plays a decisive role in environmental governance, most of them analyzed this issue from the theoretical rather than empirical perspective. Only a few empirical studies on the impact of government environmental auditing on comprehensive environmental quality have been done.

We address the aforementioned challenges in the context of China. Does government environmental auditing impact environmental quality significantly? Moreover, what are the mechanisms? Based on the data of 29 provinces/municipalities in China (except Hong Kong, Macao, Taiwan, Xizang, and Hainan) from 2004 to 2019, this paper constructed the environmental degradation index (*EDI*) to comprehensively measure environmental quality, and then conducted a theoretical analysis and empirical study on the relationship between government environmental auditing and quality of the environment. Furthermore, it explored the heterogeneous treatment effect across different groups of inter-governmental competition, financial status, and institutional environment. Finally, it conducted the robustness testing. This was performed to enrich the research on environmental governance and delineate the role of government auditing on environmental issues. A reasonable examination of the internal connection between the two may be more conducive to solving the dilemma of economic growth, resource depletion, and environmental degradation in the process of China’s development, and to provide empirical evidence for accelerating the transformation of Chinese production and lifestyle to being green, low carbon, recyclable, and sustainable.

The contributions of this paper are as follows: First, it examined the impact of government environmental auditing on the quality of the environment, which not only enriches and expands the existing research but also provides more direct and robust empirical evidence to improve the natural environment. Second, it revealed the environmental auditing mechanism and how it shapes comprehensive environmental quality. Possible heterogeneous issues were investigated and the evidence for them was found in different situations. Third, previous studies mostly used a single index to measure environmental quality. Based on what is actually happening in China, this paper selects 15 indicators from six dimensions; that is, wastewater, waste gas, industrial solid waste, air quality, domestic waste treatment, and environmental self-purification rate, to comprehensively measure environmental quality. Doing so can make up for the limitations of the existing environmental quality indicators to a certain extent. Fourth and lastly, the research findings can help local governments improve their environmental auditing responsibilities.

The rest of this paper is organized as follows. Section 2 reviews the related literature. Section 3 discusses influencing mechanisms and the research hypothesis. Section 4 describes the model, variables, and data. Section 5 reports our main empirical findings and further testing. Section 6 concludes the paper with policy implications.

## 2. Literature Review

### 2.1. The Function of Government Auditing

The existing research on government auditing mainly focuses on the definition of its function and the play of its function.

Studies on the definition of a government audit function mostly use the normative paradigm. It mainly lies in answering the question “What can government audit do?”. In 2007, Jiayi Liu, China’s former Auditor General, at the National Audit Conference first put forward the idea that the “modern government audit is an ‘immune system’ for economic and social operation”. Immune system theory points out that a government audit has the basic functions of prevention, disclosure, and defense. Subsequently, several scholars studied the immune system theory and used it to explain how the government audit function works.

Studies on the play of a government audit function are combined with the normative and empirical paradigm, and mainly focus on economic development, government governance, state-owned enterprise (SOE) governance, and so on. On the relationship between government audits and economic development, studies mainly focus on the impact of government audits on the transformation of the economic development mode and economic security, and most of them found that government audits can promote long-term economic development [22,23,24]. Studies about government audits and government governance have found that government audits can enhance government transparency, consolidate the operating efficiency of financial funds, improve government’s disclosure of financial information [25,26,27,28], ensure financial security [29,30], and discourage or prevent the corruption of officials [31,32,33,34]. In terms of government audits and SOE governance, scholars mainly found that a government audit is conducive to improving the capacity for innovation [35], quality of earnings [36], internal controls [37], investment efficiency [38,39], and promoting good development outcomes [40].

### 2.2. Factors Influencing Environmental Quality

Environmental quality has become the focus of many people’s attention, and a series of studies on the factors influencing it have been carried out in academic circles from different perspectives. Among them, the most representative is the proposal and verification of what is known as the Environmental Kuznets Curve (EKC) [41,42,43]. The EKC states that the relationship between economic growth and environmental quality presents an inverted U-shaped relationship. Subsequently, many scholars have studied other factors affecting environmental quality such as international trade [44,45,46], foreign investment [47,48,49], industrial structure [50,51,52], technological progress [53,54], income distribution [55,56,57], institutional set-up [58,59,60], and public participation [61]. However, due to the differences in economic development models and how industries are structured, there is no one model that can explain the relationship between specific factor and environmental quality, applicable to all regions and pollutants.

For the measurement of environmental quality, some scholars used a single index to evaluate, such as CO_2_ emissions [62,63] and industrial water pollution [50], and some scholars selected several specific pollutants as environmental indicators. Cole [44] chose 10 air and water pollutants; He et al. [51] selected three of the most important pollutants in the air: total suspending particles, sulfur dioxide, and nitrogen oxide; Zhang et al. [52] also selected three indicators: industrial sulfur dioxide emissions, industrial smoke (powder) dust emissions, and days with air quality above grade II; Sharif [54] used CO_2_, N_2_O, CH_4_, and ecological footprint; Feng [64] selected the total wastewater discharges, ammonia nitrogen emissions, chemical–oxygen demand (COD) emissions, total nitrogen emissions, total phosphorus emissions, sulfur dioxide emissions, oxide emissions, and soot and dust emissions as indicators; Saghaian [65] used N_2_O and CH_4_ emissions and Feng [64] selected wastewater discharges and exhaust emissions as proxies for the environmental quality. In addition, some scholars evaluated environmental quality from the perspective of carbon emissions [66,67].

Due to the complexity and interactions of environmental systems, the traditional one or several specific pollutants can only represent a specific aspect of the environment [68], and are incapable of accurately assessing the overall quality of the environment [69]. To meet the needs of a comprehensive evaluation of the multi-dimensional indicators of an environmental system, Liu et al. [56] utilized the entropy method to calculate a comprehensive index for environmental pollution from gas emissions, solid wastes, and wastewater; Luo et al. [70] proposed RSEI for an environment assessment, which consists of four coupling components: green index, wetness, dryness, and heat. Compared with a single index, the comprehensive environmental index can better reflect the overall state of the environment in a given region [71].

### 2.3. The Impact of Government Auditing on Environmental Quality

The research on the impact of government auditing on environmental quality mostly adopts the normative research paradigm, but relatively few empirical research studies have been conducted. Based on the experimental data derived from outgoing officials’ auditing of natural resources in China, Wu et al. [72] found that regional water quality had improved, but this was not the case for exhaust gas and smoke. Sun et al. [73] empirically found that outgoing officials’ auditing of natural resources could improve the quality of the environment and achieve sustainable economic development to some extent.

With data from the “Three Rivers and Three Lakes” Water Pollution Prevention and Control Performance Audit Survey, Zeng et al. [74] discovered that government environmental auditing improved the living conditions for fauna and flora, and the greater the intensity of government audit, the less strife there was between the various levels of government about how to care for the environment. Cai et al. [75] claimed that government environmental auditing did improve the level and quality of corporate information disclosure regarding environmental matters. Jiang et al. [76] found that China’s environmental auditing significantly improved the state of the environment, but only in the short-term.

Xiong et al. [77] introduced the natural resource asset accountability audit evaluation system by combining the entropy weight method and TOPSIS method, and constructed a system based on an energy subsystem, economy subsystem, and environment subsystem, and evaluated the performance of leading officials of the Jiangxi Province in China. The results indicated that the overall situation of natural resource assets showed an upward trend, and the overall performance should be recognized.

Yu et al. [78] used the data envelopment analysis (DEA) model to calculate the efficiency in treating air pollution and studied its relationship with government audits. They found that government audits did improve how air pollution was dealt with, but it did differ from different financial conditions. Xie et al. [79] found that government audits improved the governance aspects of managing environmental pollution. The greater the input of government audits, the better that supervision and consultation were, and subsequently the more conducive it was to improving the disposal efficiency of the industrial “three wastes” in the region.

Based on the quasi-natural experimental context of the leading officials’ natural resources accountability audit (NRAA), Zeng et al. [80] empirically found that after the implementation of the NRAA, heavily polluting firms were more likely to adopt a source prevention strategy than an end-of-pipe governance strategy to cope with the local government’s environmental management pressure, and the environmental governance pressure transmitted to firms differs due to different promotion expectations of local governments. Yu et al. [81] examined the impact of a government environmental audit on corporate environmental performance using the difference-in-differences method, and found that government environmental audits had a significant positive impact on corporate environmental performance, especially in companies with greater audit intensity, greater penalty power, and greater judicial power, companies with stronger government environmental supervision, media environmental supervision, and public environmental supervision.

## 3. Influencing Mechanism and Research Hypothesis

Government auditing seeks to answer the needs of national governance, and it is a system that clarifies the procedures of policy that are carried out according to the law. Government auditing plays an important role in promoting social cohesion and the rule of law, promoting the proper operation of the economy, and protecting the fundamental rights and interests of all people. The call is made for the public to “do the right thing and do the right thing” in the workplace.

The concept of entrusted economic responsibility is a widely applicable theory of audit motivation. Many scholars regard this responsibility as an important condition or primary premise for generating an audit. The public accountability is the expansion and expression of the economical responsibility in the field of public property. The essence of government auditing is the independent supervision of practices and procedures, essentially to ensure the comprehensive, proper, and effective performance of government-related tasks. The development of the economy promotes public entrusted economic responsibility, and the roles and functions of government auditing also expand and evolve accordingly. Global climate change, ecosystem degradation, and the frequent occurrence of catastrophic climate-related events not only endanger economic progress but also directly threaten the survival and development of all mankind, which makes environmental issues hugely important. When there is a great contradiction between the environment and economic development, public entrusted economic responsibility also changes and what emerged is greatly expanded government auditing of environmental matters.

Government auditing, with its deterrent effect and advantages of independence, objectivity, and justice, can give a warning about the risks and hidden dangers in economic and social issues, and promote better policies and measures. On one hand, through auditing the formulation and implementation of environment-related policies and measures (including changing how industries are structured and setting emissions requirements), audit institutions can improve the implementation and content of policies. On the other hand, through auditing the raising, distribution, management, and use of environmental protection funds, the government environmental auditing can promote the rational allocation of financial funds, thus promoting the improvement of environmental quality. Moreover, through effective auditing activities, audit institutions can force local governments to fulfill their environmental responsibilities and protect the remaining natural resources, thus protecting the natural world as best they can. Based on this, we propose research hypothesis H_1_.

**Hypothesis** **1.***Governmental environmental auditing can help improve environmental quality*.

Since the end of the 20th century, the topic of “environment and economy” has been hotly debated, especially through the prism of the “Environmental Kuznets Curve” (EKC) [41,42,43,82]. China’s fiscal decentralization system effectively solves the incentive problem of local governments. A local government appraisal system with relative performance as the core is widespread [83], which gives rise to the “scale competition” among regional governments under the relative performance scale [84].

Although the relative performance appraisal system has effectively promoted the growth of China’s economy [85], it has ignored the state of the resources and environment, resulting in the excessive use of natural resources and the rapid increase in the amount of waste produced. To improve environmental quality, the report of the 19th National Congress of the Communist Party of China (CPC) in 2017 officially approved and implemented an audit system to check local officials’ management of resource assets. However, despite the introduction of this audit policy, its effect on the environment was minimal at best and there was not enough time to implement it. Moreover, under the administrative audit mode and dual leadership system in China, the personnel appointment and removal and fund allocation of local audit institutions are mainly in the charge of the people’s government at the same level, which may affect the independence of local audit institutions. Independence and integrity are the hallmarks of an audit, and any undermining of them will directly compromise what the audit function seeks to do. Given that China’s regions compete over economic development, local governments pay more attention to economic growth and will tend to intervene in auditing measures, so the efficiency of local audit institutions is affected leading to a worse-off environment. Based on this, we propose research hypothesis H_2_.

**Hypothesis** **2.***The smaller the government competition is, the better the effect will be of government environmental auditing on improving the quality of the environment*.

The funds needed by local audit institutions in China are mainly responsible and made available by the people’s government at the same level and included in its budget. Generally speaking, the better the financial situation of a given region, the stronger the professional competence of the auditors and the better the audit efficiency will be. Moreover, the regions that are better off financially will have more government support for their auditing processes, such as the application of big data technology, which is more conducive to the improvement and promotion of audit ability, audit quality and audit efficiency, greatly expand the breadth and depth of audit supervision, and make the audit function play better. Based on this, we propose research hypothesis H_3_.

**Hypothesis** **3.***The better the financial status of a local government, then the more significant the effect of government environmental auditing on environmental quality improvement*.

The institutional environment wields a key influence on organizational performance [86]. However, government environmental auditing has two contrary effects on environmental quality. On one hand, regions with a better institutional environment have a much better rule of law, a more coherent and cohesive social and economic order, and proper governance procedures and processes in place. As a result, more attention and supervision are paid to the performance of public officials entrusted with responsibility in these regions, as these leaders can restrain opportunism in the wielding of public sector authority. So, these regions usually have less government intervention and higher auditing independence. In this way, the better the institutional environment is, the higher the role of government environmental auditing is, and subsequently the environment will have recovered better. Conversely, the better the institutional environment is, the more perfect the development of market intermediary organizations and the legal system environment are, and the more fully the role of market mechanism is exerted. Thus, the public will depend less on government public accountability through government auditing, and the role of the government is diminished. Therefore, the better the institutional environment is, the lower the government’s environmental auditing function, and the less significant its effect on environmental quality improvement. Based on the above analysis, this paper proposes research hypotheses H_4a_ and H_4b_.

**Hypothesis** **4a.***The better the institutional environment is, the more significant the effect of government environmental auditing will be on improving the quality of the environment*.

**Hypothesis** **4b.***The weaker the institutional environment is, the more significant the effect of government environmental auditing will be on improving the quality of the environment*.

## 4. Model, Variables, and Data

### 4.1. Sample Selection and Data Sources

Data on government environmental auditing originate from *the China Audit Yearbook.* Due to delays in its publication, the yearbook published in 2021 contains the data for 2019. We selected the panel data of 29 provinces (Tibet and Hainan were excluded due to the lack of environmental quality data. Data for Xinjiang are the sum of the Xinjiang Uygur Autonomous Region and the Xinjiang Construction Corps), autonomous regions, and municipalities from 2004 to 2019 as the sample for this research. After deleting missing variables, a total of 464 sample observations (the audit penalty data began in 2006, and when it serves to measure government environmental auditing, the sample size is 406. The data on audit reports and special audit survey reports began in 2007, and when these indices are used to measure government environmental auditing, there are 377 observations) were obtained. The data on environmental quality is from the China Statistical Yearbook and China Environmental Yearbook*,* and this was gathered through manual sorting and processing. Marketization index data are from China’s Provincial Marketization Index Database. The data concerning inter-governmental competition (competition between China’s regional governments), financial situation, and other control variables are from the official website of the National Bureau of Statistics and China Statistical Yearbook. To avoid the influence of extreme values on the research results, all continuous variables are indented by up and down 1%. The data processing software used is Stata13.

Based on the actual situation in China, we constructed the environmental degradation index (*EDI*) to comprehensively measure environmental quality. The *EDI* is constructed from six dimensions: wastewater, waste gas, industrial solid waste, air quality, solid waste treatment, and environmental self-purification rate, in total including 15 indices (see a description in Appendix A). Formulas (1)–(4) show how the *EDI* is calculated.
(1)$$EX_{ij}=(X_{ij}-X_{i}^{*})/(X_{i}^{**}-X_{i}^{*})$$
(2)$$\mathrm{Or}:EX_{ij}=(X_{i}^{**}-X_{ij})/(X_{i}^{**}-X_{i}^{*})$$
(3)$$EDXI_{ij}=1/n\sum_{i=1}^{n}EX_{ij}$$
(4)$$EDI_{j}=1/6\sum_{i=1}^{6}EDIX_{ij}$$

*EX_ij,_ EDIX_ij_,* and *EDI_j_* represent the indicator value of dimension *i*, the value of dimension *i*, and the *EDI* of province (city) *j*, respectively. The smaller the *EDI_j_* is, then the less environmental pollution there will be and subsequently environmental quality will be better.

Most existing empirical studies on government environmental auditing are based on the auditing data of central government-owned enterprises, while the empirical literature at the local government level is relatively scarce, and most of them use overall indicators of government auditing to measure auditing of environmental issues. This paper measures the performance of government environmental auditing from three aspects: audit power *(Apower*), audit execution strength *(Apush*), and audit information disclosure (*Areport*).

*Apower.* Auditing staff are not only the core but also the most critical component of audit power. The number and quality of auditors determine the strength of the audit power. When the good quality and number of auditors is assured, then the more credible their reports will be. Therefore, this paper selects the number of staff employed in audit institutions in each province as the measurement index of audit power. The larger the index then the stronger the environmental auditing power of the province’s government.

*Apush*. It is difficult to obtain specific information on the audit implementation; the government audit has taken the audit penalty amount as an important result of the audit practice for a long time. The greater the audit penalty, the more amounts of problems found by the government audit through supervision. Therefore, this paper uses the amounts of audit penalty imposed to measure the effectiveness of audit execution. The larger the audit penalty imposed, then the better the audit execution is.

*Areport.* It refers to the degree to which audit information is provided. External supervision is an important way for a government audit to play its role, while the carrier of external supervision to play its role is the publicly disclosed audit information. The more sufficient information disclosed, the stronger the role of external supervision can play, and then the better the function of the government audit. This paper uses the number of audit reports and special audit investigation reports to measure the degree of how much is disclosed. The larger the index is, the better the information disclosure is, and the better the function of government audit is played.

### 4.2. Model Setting

Based on published studies [42,43,44], we estimate the following model to test our research hypothesis:(5)$$\begin{array}{l} EDI=\alpha_{0}+\alpha_{1}A\mathrm{udit}+\alpha_{2}GDP+\alpha_{3}Indstr+\alpha_{4}RD+\alpha_{5}Envginv\\ +\alpha_{6}Envginvr+\alpha_{7}Forcapr+year+prov+\varepsilon \end{array}$$
where *EDI* is the environmental degradation index and audit denotes the indicators for government environmental auditing, which include *Apower*, *Apush,* and *Areport*. To ensure the robustness of the conclusions, we controlled the following variables that might have an impact on environmental quality.

(1)GDP per capita (*GDP*). Many studies have shown that environmental quality is closely related to economic development. The EKC theory holds that there is an inverted U-shaped relationship between economic growth and environmental pollution [41,42,43]. This paper uses GDP per capita to control the economic growth.(2)Industrial structure *(Indstr)*. Over the past decades, the rapid development of secondary industries has had a beneficial effect on China’s economic growth. However, it has also brought about serious environmental problems and pollution emissions. Industrial structure also reflects the status of industrial distribution. This paper applies the ratio of secondary industry output value to total output value as the industrial structure indicator [59].(3)Technological progress (*RD*). Technological progress is closely related to environmental quality. Many studies found that technological progress can effectively alleviate environmental pollution, and it might be an important driving force to reduce the pollution emissions [53]. This paper uses the natural logarithm of R&D expenditure of the whole society as the *RD* indicator.(4)Environmental pollution control *(Envginv, Envginvr).* Environmental pollution control is an important factor affecting environmental quality. The more investment in environmental governance, the better the improvement of the environment, and the higher the environmental quality. Due to there being certain differences among local governments in economic development, local environment, industrial distribution, and other aspects in China, only absolute variables to measure the local environmental governance efforts are used, and there will be some regional deviation. The relative indicators standardize the differences between regions, which is more conducive to a horizontal comparison between regions. Therefore, this paper measures the treatment of environmental pollution from absolute and relative numbers, respectively.(5)Foreign capital *(Forcapr).* Foreign direct investment has a significant impact on environmental quality. Some studies analyzed the relationship between the two, and proposed the “Pollution Haven” hypothesis, while some research argued that foreign investment would have a positive impact on environmental quality, namely the “Pollution Halo” hypothesis [45,46,47,48,49]. Studies on its relationship have not yet reached a consensus, so foreign direct investment should be considered as one of the control variables to study environmental quality.

In addition, year and province effects are controlled. All the regression analyses adopt robust adjusted standard errors. The variables are defined in Table 1.

## 5. Empirical Results

### 5.1. Descriptive Statistics

Table 2 reports the descriptive statistics (in order to make a statistical analysis of the original variables of the government environmental audit, the descriptive statistics in Table 2 did not carry out logarithmic processing on the variables) which refer to government environmental auditing. The mean value of audit power (*Apower*) is 1015, meaning that the average number of auditors in local government agencies is 1015. The minimum number of auditors in local audit offices is 289 and the maximum number is 1863, suggesting that the number of auditors in the audit offices fluctuates greatly. The average audit penalty (*Apush*) for local auditors is CNY 29.72 billion, and the minimum and maximum values are CNY 0.16 billion and CNY 187.00 billion, respectively, indicating that the difference in the amounts paid is also very large. The average number of audit reports and special audit investigation reports (*Areport*) issued is 5053, while the minimum and maximum values are 466 and 15,703, respectively. The three indicators above all reveal significant differences between China’s provinces in terms of government environmental auditing.

Figure 1 shows the annual trend shared by government environmental auditing and environmental pollution from 2004 to 2019. The three indicators of government environmental auditing—*Apower, Apush*, and *Areport*—indicate virtually the same increase between 2004 and 2019. The environmental pollution variables as a whole show a decreasing trend which contradicts the trend of government environmental auditing. Specifically, between 2007 and 2013, government environmental auditing rose while the environmental pollution indicators fell sharply; between 2013 and 2014, government environmental auditing fell while the environmental pollution indicators rose; between 2014 and 2016, government environmental auditing rose slowly while environmental pollution fell; and between 2016 and 2019, government environmental auditing fell while environmental pollution shows a trend of rising, falling, and rising again. The opposing trends that existed between government environmental auditing and environmental pollution, to some extent, explain the important role played by government auditing in reducing environmental pollution.

Table 3 reports the descriptive statistics. It emerges that the *EDI* has a mean value of 0.417, with a minimum and maximum value of 0.166 and 0.719, respectively. The mean value of *FD* is 0.028, with a minimum value of 0.004 and a maximum value of 0.074. The mean value of *Finsit* is 6.962, with minimum and maximum values being 4.449 and 8.745, respectively. The minimum and maximum values of *MI_mea* are −4.183 and 3.516, respectively, indicating there is a large difference in the level of marketization among the samples. The mean values of *GDP*, *Indstr*, *RD*, *Envginv, Envginvr,* and *Forcapr* are 10.37, 46.38, 5.006, 4.975, 1.324, and 2.184, respectively. It is also known from the table that there are large differences between the minimum and maximum values of the control variables. This shows that there is some imbalance in economic development, industrial layout, R&D investment, pollution control, and foreign investment in China. This lays a suitable data foundation for the research in this paper.

### 5.2. Empirical Results

Table 4 summarizes the univariate test and correlation coefficients, where Panel A shows the univariate test and Panel B has the correlation coefficient. The groups in Panel A are divided by the median of *Apower, Apush,* and *Areport*, followed by the mean and median tests. Panel A suggests that environmental quality in the *Apower*, *Apush,* and *Areport* group is significantly better than in the other group (*p*_value = 1%). It means that government environmental auditing can significantly improve environmental quality, which tentatively verifies hypothesis H_1_. In Panel B, the correlation coefficients between *Apower, Apush, Areport,* and *EDI* are all negative (*p* = 1%, 1%, and 5%, respectively), indicating that government environmental auditing can significantly control environmental pollution, further verifying hypothesis H_1_.

In the VIF test of Panel C, the mean value of VIF was 4.34, and the VIF value of each explanatory variable did not exceed 10, which was strictly less than 5 except for *RD* and *Envginv*, indicating that there was no multicollinearity problem between the variables.

Table 5 encapsulates the basic regression results. Columns (1)–(3) present the regression results of *Apower, Apush,* and *Areport*, respectively. The coefficient of *Apower* is −0.124 (*t*−value = −3.42, *p* = 1%), suggesting that when government environmental auditing has more authority, there will be less environmental pollution and better environmental quality. Auditors are the most important part of the government’s audit strength. Existing studies take the number of auditors as an alternative index of the audit strength of audit institutions. Therefore, the more auditors there are, the more powerful the audit force, and the more problems in the ecological environment, so as to achieve the effect of deviation and improve the comprehensive quality of the environment. The coefficients of *Apush* and *Areport* are negative at the 10% and 5% significance level, respectively, which means that superior government environmental auditing implementation and information disclosure will lead to less environmental pollution and better environmental quality. The audit that found the problems and dealt with the punishment can better reflect the implementation of the government audit and the fund is one of the important carriers where the government audit plays a role. The disclosure of audit information is an important way for external supervision to play a role. The more sufficient the disclosure of environmental audit results, the more conducive it is to the function of government environmental audit. These results fully support hypothesis H_1_ which asserts that the function of government environmental auditing will improve environmental quality.

Table 6 presents the subgroup regression results with grouping based on inter-governmental competition. The sample was divided into lower and higher inter-governmental competition groups based on the median of inter-governmental competition (*FD*) (the sample sizes of the government auditing variables were not consistent, so they were grouped separately based on corresponding medians. The same grouping was used later in the paper). Columns (1)–(3) report the regression results of the lower group and columns (4)–(6) report the results concerning the higher one. It can be seen that, only in the lower group, *Apower, Apush,* and *Areport* are significantly negatively correlated with *EDI*. This indicates that when there is less rivalry between regional governments, the more significant will be the effect of government environmental auditing on environmental pollution, which validates hypothesis H_2_. In regions with higher local government competition, local governments have greater promotion pressure. In order to successfully obtain political promotion, local officials will strengthen their attention to the local economic development, and then conduct a timely administrative intervention in the government audit to reduce the audit function. Therefore, in the regions where the local government competition is fierce, the effect of a government environmental audit on improving environmental quality is not obvious.

Table 7 shows the subgroup regression results where the grouping is based on the fiscal situation (*Finsit)*. The sample was divided into better and worse financial situation groups based on the median of *Finsit*. Columns (1)–(3) and (4)–(6) present the regression results for the better and worse groups, respectively. In the better group, the coefficients between *Apower, Apush*, and *Areport* and *EDI* are significantly negative, which means that government environmental auditing strongly improves environmental quality when the financial situation is better, thus supporting hypothesis H_3_. When the regional financial situation is better, the local governments have more financial and material resources to invest in environmental governance and environmental audit. Human resources and property are important guarantee for the government to play the audit function. The more sufficient the funds, the stronger the professional competence of the personnel, and the more conducive to the play of the audit function. Therefore, in the areas with a good financial situation, the government environmental audit plays a more significant role in comprehensively improving environmental quality.

Table 8 gives the subgroup regression results with the grouping based on the institutional environment (*MI_mea*). The sample is divided into groups with a better and weaker institutional environment according to the median of *MI_mea.* Columns (1)–(3) and (4)–(6) show the regression results of the weaker and better group, respectively. The regression coefficients of *Apower, Apush,* and *Areport* are all significantly negative in the weaker group, but not in the better group. This suggests that the weaker the institutional environment is, the better that government environmental auditing will be for environmental quality, which verifies hypothesis H_4b_. The better the regional institutional environment is, the better the market intermediary organizations and legal system can develop, and government auditing will play less of a role. It can be stated that a better institutional environment reduces the need for government environmental auditing to enhance the quality of the environment. Conversely, the weaker the institutional environment is, the better the government environmental auditing is in improving overall environmental quality.

### 5.3. Robustness Test

#### 5.3.1. Re-Measuring Government Environmental Auditing

The previous analysis used the absolute indicators of government environmental auditing, which may be affected by regional differences. This paper further uses relative indicators to re-measure government environmental auditing variables. Specifically, audit power, audit implementation strength, and audit information disclosure are re-measured by: the ratio of personnel in local audit offices to those in the urban population (*Apower_r*); the ratio of the number of audit penalties to the amount of audit-detected problems (*Apush_r*); and the ratio of audit recommendations adopted to audit recommendations made (*Areport_r*). The regression results documented in Table 9 show that *Apower_r, Apush_r*, and *Areport_r* significantly negatively correlated with *EDI* (*p* = 1%, 1%, and 10%, respectively). This is consistent with the findings of the previous study and they are robust.

#### 5.3.2. Lagging Effect of Government Environmental Auditing

Studies show that the function of a government audit has a certain lagging effect. To test whether there is a certain lagging effect on the impact of environmental quality, we lagged government environmental auditing variables by one and two periods and obtained *Apower_L, Apush_L, Areport_L*, *Apower_L2, Apush_L2,* and *Areport_L2* as the explanatory variables for retesting. The results in Table 10 show that the coefficients of the variables are all negative with most of them being significant, which strongly suggests that government environmental auditing has a certain lagging effect on improving environmental quality, further validating the hypotheses devised for this paper.

## 6. Conclusions

Over the past decades, the extensive development in China has resulted in rapid economic growth while causing environmental pollution and the waste of resources. Coping with environmental degradation and promoting green transformation are goals shared by all countries worldwide. The 14th Five-Year Plan for national economic and social development adopted in 2021 clearly emphasizes the need to promote green development and promote a harmonious coexistence between human and nature, and put forward five specific binding targets and 14 major projects for green ecology. How to prevent and control pollution to improve the comprehensive quality of the environment is of great significance to the completion of the national 14th Five-Year Plan and the realization of high-quality economic development. So, based on the data of 29 provinces/municipalities in China (except Hong Kong, Macao, Taiwan, Tibet, and Hainan) during 2004–2019, this paper conducted a theoretical analysis and empirical test on the impact of government environmental auditing on green and low-carbon development.

The results are concluded as follows: (1) Government environmental auditing can improve the environmental quality, and there is a significant lag effect here. (2) When there is less rivalry between provincial governments, the effect of environmental auditing on overall environmental quality is more evident. (3) The better the financial situation is, the more significant the effect of government environmental auditing on environmental quality improvement. (4) The weaker the institutional environment is, the more evident the effect of environmental auditing on overall environmental quality.

The conclusions of this paper provided direct empirical evidence for understanding the function and role of government environmental auditing in environmental matters and their governance, and also provide a policy implication for how economic development should proceed. Based on the research conclusions of this paper, the following policy recommendations can be provided: (1) While improving the overall environmental quality when government environmental auditing is involved, we should take into account the influence of external factors such as government competition. We should especially consolidate environmental auditing in the areas with fierce government competition, poor financial status, and good institutional environment, in order to expand the scope of government environmental auditing. (2) Due to the certain lag of government environmental audits, the follow-up audit of environmental policy implementation should be carried out to ensure the sustainability of environmental audits. Continuous audits put higher requirements for the speed and accuracy of information sources, information processing, and information feedback. In order to ensure the effectiveness of continuous auditing, we should further promote the construction of informatization of resources and environment audit, and actively explore new audit modes to adapt to the diversity of resources and environment. (3) Auditors as the core of the audit function. The complex and changeable environment puts forward higher requirements for the professional competence of environmental audit staff, which requires auditors to understand the causes of all kinds of environmental pollution and the relevant professional knowledge of treatment. Therefore, the training of auditors should be strengthened. Through expert explanations, seminars, research, and exchanges, we can help auditors to have a more comprehensive understanding of the professional knowledge of resources and the environment, so as to help them reveal environmental problems more accurately, so as to improve the quality and efficiency of environmental audit.

## Figures and Tables

**Figure 1 ijerph-20-02770-f001:**
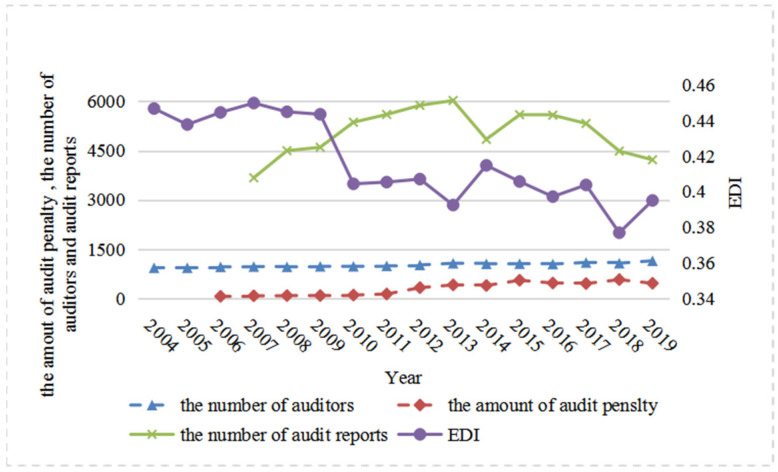
Government environmental auditing and environmental quality trends from 2004–2019.

**Table 1 ijerph-20-02770-t001:** Definitions of Variables.

	Variable Name	Symbol	Measurement Method
Environmental quality	Environmental degradation index	*EDI*	Composite environmental index
Government environmental auditing	Audit power	*Apower*	The natural logarithm of the number of personnel of the local audit institution
	Audit execution strength	*Apush*	The natural logarithm of the amount of audit penalty
	Audit information disclosure	*Areport*	The natural logarithm of the number of audit reports and special audit investigation reports
Grouping Variable	Intergovernmental competition	*FD*	The ratio between local fiscal expenditure and national fiscal expenditure
	Financial situation	*Finsit*	The ratio between general budget expenditure and general budget revenue
	Institutional environment	*MI_mea*	Marketization index (Because the marketization indexes have inconsistencies in caliber. Therefore, de-meaning is adopted in the grouping test in this paper to ensure the comparability of data)
Control variable	GDP per capita	*GDP*	The natural log of real GDP per capita based on 1996
	Industrial structure	*Indstr*	Proportion of the output value of the secondary industry in GDP
	Technological progress	*RD*	The natural logarithm of the R&D expenditure of the whole society
	Pollution control	*Envginv*	The natural logarithm of the total investment in environmental pollution control
	Pollution control	*Envginvr*	The proportion of total environmental pollution control investment in GDP
	Foreign capital utilization	*Forcapr*	The proportion of the actual utilization of foreign capital in GDP

**Table 2 ijerph-20-02770-t002:** Descriptive statistics of government environmental auditing variables.

Variables	Sample	Mean	Standard Deviation	Min	Median	Max
*Apower*	464	1015	339.7	289	992	1863
*Apush*	406	29.72	34.25	0.16	15.61	187.00
*Areport*	377	5053	3174	466	4635	15,703

**Table 3 ijerph-20-02770-t003:** Descriptive statistics.

Variables	Sample	Mean	Median	Min	25% Quantile	50% Quantile	75% Quantile	Max
*EDI*	464	0.417	0.106	0.166	0.347	0.405	0.484	0.719
*FD*	464	0.028	0.013	0.004	0.019	0.025	0.034	0.074
*Finsit*	464	6.962	0.941	4.449	6.206	7.104	7.730	8.745
*MI_mea*	464	0.000	1.733	−4.183	−1.250	0.088	1.298	3.516
*Apower*	464	6.853	0.404	5.666	6.685	6.899	7.132	7.530
*Apush*	406	14.24	1.263	9.667	13.33	14.26	15.21	16.74
*Areport*	377	8.279	0.780	6.144	7.945	8.441	8.899	9.662
*GDP*	464	10.37	0.733	8.353	9.879	10.49	10.88	11.90
*Indstr*	464	46.38	7.661	16.16	42.38	47.40	51.95	61.50
*RD*	464	5.006	1.430	1.084	4.043	5.002	6.072	8.039
*Envginv*	464	4.975	0.976	1.668	4.339	5.076	5.680	7.256
*Envginvr*	464	1.324	0.745	0.299	0.859	1.183	1.574	9.211
*Forcapr*	464	2.184	1.794	0.010	0.771	1.772	2.980	8.783

**Table 4 ijerph-20-02770-t004:** Univariate tests, correlation coefficients and VIF test.

Panel A Univariate Test							
Variables	Low Audit Force (N = 230)		High Audit Force (N = 230)		T-value	Z-value	
	Mean	Median	Mean	Median			
*EDI*	0.437	0.425	0.397	0.392	4.0873 ***	2.912 ***	
Variables	Low audit execution strength (N = 174)		High audit execution strength (N = 174)		T-value	Z-value	
	Mean	Median	Mean	Median			
*EDI*	0.432	0.422	0.395	0.382	3.5183 ***	3.339 ***	
Variables	Low audit information disclosure (N = 159)		High audit information disclosure (N = 160)		T-value	Z-value	
	Mean	Median	Mean	Median			
*EDI*	0.425	0.420	0.396	0.386	2.6257 ***	2.100 ***	
**Panel B Correlation Coefficient**							
Variables	*EDI*	*Apower*	*Apush*	*Areport*	*FD*	*Finsit*	*MI_mea*
*EDI*	1						
*Apower*	−0.38 ***	1					
*Apush*	−0.23 ***	0.52 ***	1				
*Areport*	−0.11 **	0.45 ***	0.52 ***	1			
*FD*	−0.44 ***	0.69 ***	0.41 ***	0.44 ***	1		
*Finsit*	−0.12 **	0.33 ***	0.71 ***	0.57 ***	0.26 ***	1	
*MI_mea*	−0.59 ***	0.49 ***	0.18 ***	−0.0500	0.64 ***	−0.13 ***	1
*GDP*	−0.33 ***	0.27 ***	0.40 ***	−0.26 ***	0.35 ***	0.45 ***	0.46 ***
*Indstr*	0.36 ***	−0.15 ***	−0.09 *	0.24 ***	0.0200	−0.18 ***	−0.0100
*RD*	−0.59 ***	0.69 ***	0.56 ***	0.21 ***	0.74 ***	0.45 ***	0.72 ***
*Envginv*	−0.33 ***	0.63 ***	0.64 ***	0.37 ***	0.63 ***	0.59 ***	0.46 ***
*Envginvr*	0.35 ***	−0.19 ***	−0.08 *	−0.0800	−0.27 ***	0	−0.32 ***
*Forcapr*	−0.31 ***	0.32 ***	−0.09 *	−0.19 ***	0.36 ***	−0.24 ***	0.67 ***
Variables	*GDP*	*Indstr*	*RD*	*Envginv*	*Envginvr*	*Forcapr*	
*GDP*	1						
*Indstr*	−0.27 ***	1					
*RD*	0.76 ***	−0.21 ***	1				
*Envginv*	0.70 ***	−0.08 *	0.83 ***	1			
*Envginvr*	0.0400	0.0100	−0.20 ***	0.23 ***	1		
*Forcapr*	0.32 ***	0.11**	0.44 ***	0.28 ***	−0.11 **	1	
**Panel C VIF Test**							
Variables	Mean	*GDP*	*Indstr*	*RD*	*Envginv*	*Envginvr*	*Forcapr*
VIF-value	4.34	3.67	1.4	9.86	8.15	2.91	1.7

Note: *, **, and *** represent significance levels of 10%, 5%, and 1%, respectively.

**Table 5 ijerph-20-02770-t005:** The impact of government environmental auditing on environmental quality.

Variables	(1)	(2)	(3)
	*EDI*	*EDI*	*EDI*
*Apower*	−0.124 ***		
	(−3.42)		
*Apush*		−0.007 *	
		(−1.96)	
*Areport*			−0.016 **
			(−2.06)
*GDP*	0.052 **	0.044 **	0.041 *
	(2.46)	(2.02)	(1.65)
*Indstr*	−0.000	−0.001	−0.001
	(−0.03)	(−1.22)	(−0.62)
*RD*	−0.021	0.003	0.006
	(−1.29)	(0.23)	(0.37)
*Envginv*	0.019 *	0.021 **	0.014
	(1.67)	(2.00)	(1.40)
*Envginvr*	−0.011 *	−0.010 **	−0.008
	(−1.89)	(−2.06)	(−1.52)
*Forcapr*	−0.000	−0.001	−0.001
	(−0.20)	(−0.36)	(−0.37)
*_cons*		−0.007 *	
		(−1.96)	
Year FEs	YES	YES	YES
Province FEs	YES	YES	YES
*N*	464	406	377
adj. *R*^2^	0.861	0.900	0.910

Note: *, **, and *** represent significance levels of 10%, 5%, and 1%, respectively. The same below.

**Table 6 ijerph-20-02770-t006:** Governmental environmental auditing, inter-governmental competition, and environmental quality.

Variables	Low Level of Inter-Governmental Competition			High Level of Inter-Governmental Competition		
	(1)	(2)	(3)	(4)	(5)	(6)
	*EDI*	*EDI*	*EDI*	*EDI*	*EDI*	*EDI*
*Apower*	−0.216 ***			0.047		
	(−4.16)			(0.95)		
*Apush*		−0.008 **			−0.006	
		(−2.03)			(−1.16)	
*Areport*			−0.027 **			−0.013
			(−2.44)			(−1.09)
*GDP*	−0.000	0.101 ***	−0.014	0.131 ***	−0.001	0.083 *
	(−0.01)	(2.63)	(−0.47)	(3.47)	(−0.03)	(1.78)
*Indstr*	0.001	−0.003 **	0.001	−0.002	−0.000	−0.003 **
	(0.79)	(−2.15)	(0.84)	(−1.31)	(−0.18)	(−2.14)
*RD*	−0.070 ***	0.042 *	−0.037 *	0.024	−0.036 *	0.046 *
	(−3.10)	(1.84)	(−1.74)	(1.12)	(−1.83)	(1.92)
*Envginv*	0.035 *	−0.015	0.020	−0.036 *	0.031 *	−0.013
	(1.94)	(−0.93)	(1.27)	(−1.70)	(1.84)	(−0.75)
*Envginvr*	−0.016 **	0.006	−0.009	0.018	−0.014 **	0.005
	(−2.22)	(0.47)	(−1.53)	(1.25)	(−2.21)	(0.36)
*Forcapr*	−0.005	−0.006 ***	−0.001	−0.005 **	0.001	−0.007 ***
	(−1.33)	(−3.55)	(−0.26)	(−2.32)	(0.25)	(−2.85)
*_cons*	1.765 ***	−0.802 **	0.904 ***	−1.348 ***	0.674 **	−0.921 *
	(5.11)	(−2.14)	(3.84)	(−3.28)	(2.38)	(−1.79)
Year FEs	YES	YES	YES	YES	YES	YES
Province FEs	YES	YES	YES	YES	YES	YES
*N*	232	203	189	232	203	188
adj. *R*^2^	0.843	0.882	0.899	0.892	0.915	0.916

Note: *, **, and *** represent significance levels of 10%, 5%, and 1%, respectively.

**Table 7 ijerph-20-02770-t007:** Government environmental auditing, financial status, and environmental quality.

Variables	Better Financial Status			Poor Financial Status		
	(1)	(2)	(3)	(4)	(5)	(6)
	*EDI*	*EDI*	*EDI*	*EDI*	*EDI*	*EDI*
*Apower*	−0.166 ***			−0.053		
	(−3.50)			(−1.26)		
*Apush*		−0.007 *			−0.009	
		(−1.70)			(−1.62)	
*Areport*			−0.025 **			−0.015
			(−2.14)			(−1.21)
*GDP*	−0.042	−0.009	0.006	−0.019	0.009	−0.007
	(−1.30)	(−0.25)	(0.18)	(−0.58)	(0.21)	(−0.19)
*Indstr*	−0.000	0.000	−0.001	0.002 *	−0.002	0.001
	(−0.18)	(0.29)	(−1.16)	(1.92)	(−0.88)	(0.35)
*RD*	−0.038 *	0.067 ***	−0.014	0.051 **	−0.018	0.060 **
	(−1.75)	(2.61)	(−0.59)	(2.27)	(−0.50)	(2.12)
*Envginv*	0.040 ***	0.022	0.023	0.048 **	0.034 *	0.024
	(2.73)	(1.30)	(1.60)	(2.54)	(1.97)	(1.41)
*Envginvr*	−0.017 ***	−0.014	−0.011 *	−0.034 ***	−0.015 **	−0.015
	(−3.25)	(−1.25)	(−1.95)	(−2.66)	(−2.60)	(−1.30)
*Forcapr*	0.006	−0.005 **	0.005	−0.007 ***	0.010	−0.004
	(1.15)	(−2.14)	(0.89)	(−3.49)	(1.27)	(−1.59)
*_cons*	1.985 ***	−0.030	0.593	0.404	0.432	−0.192
	(4.60)	(−0.09)	(1.65)	(0.93)	(1.04)	(−0.47)
Year FEs	YES	YES	YES	YES	YES	YES
Province FEs	YES	YES	YES	YES	YES	YES
*N*	232	203	189	232	203	188
adj. *R*^2^	0.906	0.921	0.928	0.845	0.876	0.885

Note: *, **, and *** represent significance levels of 10%, 5%, and 1%, respectively.

**Table 8 ijerph-20-02770-t008:** Government auditing, institutional environment, and environmental quality.

Variables	Weak Institutional Environment			Better Institutional Environment		
	(1)	(2)	(3)	(4)	(5)	(6)
	*EDI*	*EDI*	*EDI*	*EDI*	*EDI*	*EDI*
*Apower*	−0.225 ***			−0.080		
	(−4.35)			(−1.16)		
*Apush*		−0.007 *			−0.006	
		(−1.67)			(−0.99)	
*Areport*			−0.031 **			−0.003
			(−2.32)			(−0.28)
*GDP*	−0.023	0.032	−0.016	0.051	−0.000	0.030
	(−0.72)	(0.86)	(−0.50)	(0.55)	(−0.01)	(0.67)
*Indstr*	0.001	−0.001	0.001	−0.000	−0.000	−0.000
	(0.98)	(−0.66)	(0.61)	(−0.05)	(−0.06)	(−0.37)
*RD*	−0.045 **	0.060 **	−0.027	0.039	−0.023	0.069 ***
	(−1.99)	(2.55)	(−1.21)	(1.25)	(−1.11)	(2.99)
*Envginv*	0.015	0.007	0.019	0.018	0.030 *	0.004
	(0.72)	(0.44)	(1.13)	(0.58)	(1.71)	(0.28)
*Envginvr*	−0.011	−0.002	−0.010	−0.015	−0.014 **	−0.001
	(−1.51)	(−0.20)	(−1.52)	(−0.65)	(−2.19)	(−0.12)
*Forcapr*	0.010	−0.005 **	0.005	−0.006 **	0.010	−0.004 *
	(1.31)	(−2.49)	(0.65)	(−2.29)	(1.53)	(−1.73)
*_cons*	1.998 ***	−0.568	0.940 ***	0.106	0.654 **	−0.437
	(5.03)	(−1.40)	(3.45)	(0.11)	(2.55)	(−1.05)
Year FEs	YES	YES	YES	YES	YES	YES
Province FEs	YES	YES	YES	YES	YES	YES
*N*	232	203	188	232	203	189
adj. *R*^2^	0.825	0.863	0.881	0.846	0.875	0.884

Note: *, **, and *** represent significance levels of 10%, 5%, and 1%, respectively.

**Table 9 ijerph-20-02770-t009:** Replacing indicators for government environmental auditing.

Variables	(1)	(2)	(3)
	*EDI*	*EDI*	*EDI*
*Apower_r*	−0.228 ***		
	(−3.17)		
*Apush_r*		−7.515 ***	
		(−2.72)	
*Areport_r*			−9.992 *
			(−1.80)
*GDP*	0.010	0.015	0.021
	(0.40)	(0.64)	(0.78)
*Indstr*	−0.000	−0.001	−0.001
	(−0.09)	(−1.14)	(−0.75)
*RD*	−0.012	0.009	0.015
	(−0.82)	(0.68)	(1.02)
*Envginv*	0.019 *	0.022 **	0.018 *
	(1.74)	(2.36)	(1.84)
*Envginvr*	−0.011 **	−0.011 **	−0.009 *
	(−2.15)	(−2.34)	(−1.93)
*Forcapr*	−0.001	−0.001	−0.001
	(−0.55)	(−0.40)	(−0.56)
*_cons*	0.215	−0.159	−0.255
	(0.92)	(−0.60)	(−0.87)
Year FEs	YES	YES	YES
Province FEs	YES	YES	YES
*N*	464	406	377
adj. *R*^2^	0.863	0.903	0.911

Note: *, **, and *** represent significance levels of 10%, 5%, and 1%, respectively.

**Table 10 ijerph-20-02770-t010:** Lagging effect of government environmental auditing.

Variables	(1)	(2)	(3)	(4)	(5)	(6)
	*EDI*	*EDI*	*EDI*	*EDI*	*EDI*	*EDI*
*Apower_L*	−0.091 ***					
	(−2.83)					
*Apower_L2*		−0.058 *				
		(−1.83)				
*Apush_L*			−0.007 **			
			(−2.13)			
*Apush_L2*				−0.007 **		
				(−2.12)		
*Areport_L*					−0.023 ***	
					(−3.01)	
*Areport_L2*						−0.014 *
						(−1.80)
*GDP*	0.043 **	0.044 **	0.046 *	0.041	0.045	0.055
	(2.13)	(2.03)	(1.81)	(1.27)	(1.46)	(1.54)
*Indstr*	−0.001	−0.001	−0.001	−0.001	−0.000	−0.001
	(−0.73)	(−1.18)	(−0.87)	(−0.83)	(−0.46)	(−0.47)
*RD*	−0.011	0.001	0.009	0.023	0.014	0.020
	(−0.76)	(0.09)	(0.56)	(1.44)	(0.86)	(1.14)
*Envginv*	0.024 **	0.023 **	0.016	0.012	0.008	0.004
	(2.31)	(2.20)	(1.59)	(1.26)	(0.84)	(0.40)
*Envginvr*	−0.012 **	−0.012 **	−0.009 *	−0.007	−0.005	−0.004
	(−2.21)	(−2.12)	(−1.75)	(−1.61)	(−1.25)	(−0.91)
*Forcapr*	−0.001	−0.001	−0.001	−0.001	−0.001	−0.001
	(−0.50)	(−0.39)	(−0.53)	(−0.68)	(−0.32)	(−0.55)
*_cons*	0.319	−0.015	−0.390	−0.427	−0.333	−0.530
	(0.99)	(−0.04)	(−1.45)	(−1.26)	(−1.03)	(−1.43)
Year FEs	YES	YES	YES	YES	YES	YES
Province FEs	YES	YES	YES	YES	YES	YES
*N*	435	406	377	348	348	319
adj. *R*^2^	0.886	0.900	0.910	0.924	0.925	0.931

Note: *, **, and *** represent significance levels of 10%, 5%, and 1%, respectively.

## Data Availability

The datasets used in this study are available from the corresponding author on reasonable request.

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
