# Peer review of "Can Government Environmental Auditing Help to Improve Environmental Quality? Evidence from China"

_ijerph, 2023, doi:10.3390/ijerph20042770_

Round 1

Reviewer 1 Report

This article examines the relationship between government environmental auditing and environmental quality, then uses heterogeneity tests for further analysis. The article is well organized and has valid conclusions, but there are still several problems.

1. The proportion of relevant literature in the last three years needs to be increased.

2. Please outline the questions that this paper seeks to investigate and subsequent arrangement of the article in a separate paragraph in the first part.

3. There are some places in this paper where the corresponding references should be cited as support. The following closely studies should be cited in your paper: Local government environmental regulatory pressures and corporate environmental strategies: Evidence from natural resource accountability audits in China; Does vertical supervision promote regional green transformation? Evidence from Central Environmental Protection Inspection; etc.

4. I believe that since the main contribution of the article mentions the selection of 15 indicators from six dimensions to measure environmental quality comprehensively, it is necessary to add a paragraph in Section 2 to describe previous scholars' measurement and measurement methods of environmental quality.

5. Please give reasons or justifications for the choice of independent variables and control variables and list the corresponding references.

6. This paper is a more in-depth study of the relationship between government environmental auditing and environmental quality. To ensure the integrity of the article, I think the authors should list the conclusions in Section 6 and provide corresponding policy recommendations based on the conclusions.

7. I believe that the green and low-carbon development mentioned in the title of this paper and the explanatory variable environmental quality are closely related but not equivalent, which may cause misunderstanding to the readers. I hope the authors will carefully consider the title of this paper.

Author Response

Dear Reviewer,

Thank you for your time spent in reading and reviewing our paper and providing these comments for us. We have made every effort to address these issues. Please see the new version of the manuscript and read our point-by-point responses for details in the “Response document”.

Thank you for your work.

Best wishes,

Yours sincerely,

Chao Feng, Ph.D.

Reviewer 2 Report

This is a well-crafted manuscript, dealing with an extremely interesting topic; data and statistical analyses are correct and results consistent with the proposed hypotheses. While I am positive about the paper, I have a major concern and some minor issues to raise.

My major concern is about the overall contextualization of the paper that should be better qualified in terms of the institutional Chinese environment, the role of the State and politics in the Chinese economy, the microeconomic Chinese structure and the role of Chinese firm-level strategies and performances (with regard both economic growth and environmental damage). This overall framework will strength Sections 1 and 2 and will be extremely useful to propose an interpretation of the main results and the ones concerning the control variables (in the current version, results from the control variables are not adequately discussed). To fill these gaps, the following references should be discussed:

Ø  Bao, Z. (2016). Innovative behavior and the Chinese enterprise survival risk: An empirical research. China Finance and Economic Review, 4, 18.

Ø  Chen, C. J., Li, Z., Su, X., & Sun, Z. (2011). Rent-seeking incentives, corporate political connections, and the control structure of private firms: Chinese evidence. Journal of Corporate Finance, 17(2), 229–243.

Ø  Fan, J. P., Wong, T. J., & Zhang, T. (2007). Politically connected CEOs, corporate governance, and Post-IPO performance of China’s newly partially privatized firms. Journal of Financial Economics, 84(2), 330–357.

Ø  Haveman, H. A., Jia, N., Shi, J., & Wang, Y. (2017). The dynamics of political embeddedness in China. Administrative Science Quarterly, 62(1), 67–104.

Ø  Khan, F.U., Zhang, J., Dong, N., Usman, M., Ullah, S. & Ali, S. (2021). Does privatization matter for corporate social responsibility? Evidence from China. Eurasian Business Review, 11, 497–515.

Ø  Li, H., Meng, L., Wang, Q., & Zhou, L.-A. (2008). Political connections, financing and firm performance: Evidence from Chinese private firms. Journal of Development Economics, 87(2), 283–299.

Ø  Liu, X., & Anbumozhi, V. (2009). Determinant factors of corporate environmental information disclosure: An empirical study of Chinese listed companies. Journal of Cleaner Production, 17(6), 593–600.

Ø  Rauf, F., Voinea, C. L., Bin Azam Hashmi, H., & Fratostiteanu, C. (2020). Moderating effect of political embeddedness on the relationship between resources base and quality of CSR disclosure in China. Sustainability, 12(8), 3323.

Ø  Rauf, F., Voinea, C.L., Roijakkers, N., Naveed, K, Hashmi, H.B.A., Rani, T. (2022). How executive turnover influences the quality of corporate social responsibility disclosure? Moderating role of political embeddedness: evidence from China, Eurasian Business Review, 12, 527–551.

Ø  Wang, Z., Reimsbach, D., & Braam, G. (2018). Political embeddedness and the diffusion of corporate social responsibility practices in China: A trade-off between financial and CSR performance? Journal of Cleaner Production, 198, 1185–1197.

Ø  Wu, W., Wu, C., & Rui, O. M. (2012a). Ownership and the value of political connections: Evidence from China. European Financial Management, 18(4), 695–729.

Ø  Yu, Z., Xiao, X. (2022). Shadow banking contraction and innovation efficiency of tech-based SMEs-based on the implementation of China’s New Asset Management Regulation. Eurasian Business Review, 12, 251–275.

Ø  Zeng, S., Xu, X., Dong, Z., & Tam, V. W. (2010). Towards corporate environmental information disclosure: An empirical study in China. Journal of Cleaner Production, 18(12), 1142–1148.

Ø  Zhang, M., Mohnen, P. (2022). R&D, innovation and firm survival in Chinese manufacturing, 2000–2006. Eurasian Business Review, 12, 59–95.

Ø  Zhu, J., Jiang, D., Shen, Y., & Shen, Y. (2020). Does regional air quality affect executive turnover at listed companies in China? Economic Modelling.

Minor issues:

1.      Please check the 1:1 correspondence between citations and references

2.      What you mean by “social” R&D in Table 1

3.      Qualify why and where Envginv is different from Envginvr in Table 1 and in the etxt

4.      How do you justify the insignificance of the regressor Indstr? A significant positive impact should emerge.

Author Response

(The authors gave the same response as above.)

Reviewer 3 Report

This work aims to examine the impact and mechanisms of government environmental auditing on environmental quality, evidence from China. The research is of great significance to provide empirical evidence for understanding the role and function of government environmental auditing in environmental governance. Personally, I think the authors have done a lot of work, however, there are some deficiencies that need to be further improved.

Comment 1: The explanatory variable in the empirical of this paper is the environmental degradation index (EDI), which is a measure of environment quality. This is not consistent with the focus of green and low-carbon development in the title. It is recommended to make a small change to the title, which is more in line with the content of this work.

Comment 2: Literature review is too much, the focus is not prominent, the logic is not clear.

Comment 3: 5.2 Empirical results, this part needs to consider the collinearity problem and give the VIF of each variable in the calculation model.

Comment 4: This work lacks the discussion part. This part need to combine with local policies, which will help to implement them. In the meanwhile, it needs to be closely combined with the previous conclusions.

Author Response

(The authors gave the same response as above.)

Round 2

Reviewer 2 Report

The Authors have implemented all the revisions requested

Author Response

Dear Reviewer,

Thank you again for your time spent in reading and reviewing our paper and providing these comments for us. We have made every effort to address these issues. Please see the new version of the manuscript and read our point-by-point responses for details in the “Response document”.

Thank you for your work.

Best wishes,

Yours sincerely,

Chao Feng, Ph.D.

Reviewer 3 Report

This version is significantly improved compared with the previous version. However, there are still some details to be improved. For example, (1) square symbol appears above the number (Line 320); (2) the figure 1 can be further improved. The authors may modify and improve the legend and the lines inside (Line 438); (3) the font format is not very consistent. Some words are italicized and some are not (Line 447, et al.); (4) for the same point of view in the paper, the references (the number is 3 to 5 may be appropriate) may be appropriately reduced (Line 152, et al.). Therefore, it is necessary to check the full text again before it accept.

Author Response

(The authors gave the same response as above.)
